# Mathematical modelling of the impact of expanding levels of malaria control interventions on *Plasmodium vivax*

Michael T. White[1], Patrick Walker[2], Stephan Karl[3,4,5], Manuel W. Hetzel [6,7], Tim Freeman[8], Andreea Waltmann[4,5], Moses Laman[3], Leanne J. Robinson[3,4,5,9], Azra Ghani[2] & Ivo Mueller[1,4,5]

*Plasmodium vivax* poses unique challenges for malaria control and elimination, notably the potential for relapses to maintain transmission in the face of drug-based treatment and vector control strategies. We developed an individual-based mathematical model of *P. vivax* transmission calibrated to epidemiological data from Papua New Guinea (PNG). In many settings in PNG, increasing bed net coverage is predicted to reduce transmission to less than 0.1% prevalence by light microscopy, however there is substantial risk of rebounds in transmission if interventions are removed prematurely. In several high transmission settings, model simulations predict that combinations of existing interventions are not sufficient to interrupt *P. vivax* transmission. This analysis highlights the potential options for the future of *P. vivax* control: maintaining existing public health gains by keeping transmission suppressed through indefinite distribution of interventions; or continued development of strategies based on existing and new interventions to push for further reduction and towards elimination.

[1] Malaria: Parasites and Hosts, Department of Parasites and Insect Vectors, Institut Pasteur, 25-28 Rue du Dr Roux, 75015 Paris, France. [2] MRC Centre for Outbreak Analysis and Modelling, Department of Infectious Disease Epidemiology, Imperial College London, London Norfolk Place, W2 1PG, UK. [3] Vector-borne Diseases Unit, Papua New Guinea Institute of Medical Research, Madang 511, Papua New Guinea. [4] Division of Population Health and Immunity, Walter and Eliza Hall Institute of Medical Research, Melbourne, VIC 3052, Australia. [5] Department of Medical Biology, Melbourne University, Melbourne, VIC 3052, Australia. [6] Swiss Tropical and Public Health Institute, Socinstrasse 57, 4051 Basel, Switzerland. [7] University of Basel, Petersplatz 1, 4001 Basel, Switzerland. [8] Rotarians Against Malaria, Port Moresby 121, Papua New Guinea. [9] Burnet Institute, Melbourne, VIC 3004, Australia. Correspondence and requests for materials should be addressed to M.T.W. (email: michael.white@pasteur.fr)

Renewed scientific and financial commitment to controlling and eliminating malaria has led to reductions in global malaria cases by 37% and reductions in malaria associated deaths by 60% since 2000[1]. Although the greatest public health burden is attributable to *Plasmodium falciparum*, predominantly in children and pregnant women in sub-Saharan Africa, it is increasingly recognised that *P. vivax* may pose a greater challenge to malaria elimination efforts[2]. For countries with co-endemic *P. falciparum* and *P. vivax*, it has been repeatedly observed that as the total malaria burden decreases, the proportion of cases attributable to *P. vivax* increases[3]. The major obstacle to *P. vivax* elimination is its ability to relapse from dormant liver-stage hypnozoites, weeks to years after clearance of the primary blood-stage infection[4].

The highest levels of *P. vivax* transmission in the world are found in Papua New Guinea (PNG)[5], with parasite prevalence by light microscopy ($PvPR_{LM}$) in excess of 10% frequently reported[6]. PNG has a wide diversity of *P. vivax* transmission settings, with large variations in altitude, many different mosquito species, and varying levels of co-endemicity with *P. falciparum*[7]. PNG has also made some of the most significant contributions to the global understanding of *P. vivax* transmission and control, ranging from clinical trials to assess the effectiveness of artemisinin combination therapies (ACT)[8,9], to the demonstration that ~80% of new blood-stage infections are attributable to relapses[10]. PNG therefore provides an excellent case study for modelling the impact of combinations of interventions on *P. vivax* across a wide range of transmission settings, providing valuable lessons that can be applied in any *P. vivax* endemic setting.

The two pillars of malaria control in PNG are treatment of symptomatic cases and vector control. National treatment protocols recommend first-line treatment of uncomplicated malaria due to all species of malaria with artemether-lumefantrine (AL), plus primaquine for cases that are positive for either *P. vivax* or *P. ovale*. Access to rapid diagnostic tests (RDTs) and treatment with AL are available in ~50% of first-line health centres or aid posts, with primaquine availability being much more limited[11]. From 2005 the PNG National Malaria Control Programme, supported by the Global Fund, has overseen nationwide, free distribution of long-lasting insecticidal nets (LLIN)[12]. These LLIN distributions have coincided with major reductions in malaria transmission, with *P. falciparum* prevalence by light microscopy ($PfPR_{LM}$) and $PvPR_{LM}$ reducing nationwide by 20–99% between 2009 and 2014, with *P. vivax* initially decreasing more slowly than *P. falciparum*[13,14]. The relative contributions made by LLINs and ACTs in the reduction of malaria prevalence in PNG are unknown, but it is likely that LLINs are the larger contributor, in line with findings from an analysis of *P. falciparum* in Africa[15]. ACTs are very effective for treating uncomplicated and severe episodes of *P. vivax*[16], but are unlikely to cause substantial reductions in onward transmission unless treatment is accompanied with primaquine to prevent relapses.

It is not possible to directly detect *P. vivax* hypnozoites with existing technology, and they can be cleared by treatment with just one class of drug, the 8-aminoquinolines (8-AQ). Primaquine is the only licensed 8-AQ, but it requires a lengthy treatment regimen and risks causing episodes of severe haemolysis in glucose-6-phosphate-dehydrogenase (G6PD) deficient individuals[17,18]. To mitigate against this risk, it has been advised that primaquine be delivered alongside testing for G6PD deficiency[19]. Screening for G6PD deficiency is not routinely available in developing countries such as PNG, although point-of-care G6PD diagnostics are becoming available in some settings[20]. The effectiveness of primaquine is often diminished because of challenges in adhering to long treatment regimens[21] and poor efficacy in individuals with low CYP2D6 metabolizer phenotypes[22].

Another 8-AQ, tafenoquine, is currently undergoing clinical trials and promises to have comparable efficacy with a single dose[23]. However, tafenoquine may also cause haemolysis in G6PD deficient individuals, so its use will be contingent on effective G6PD screening.

Although there is evidence for the role played by vector control in reducing levels of *P. vivax*, there is no strong evidence base for the potential impact of widespread treatment with primaquine or tafenoquine on reducing population-level transmission in countries with high levels of transmission of tropical strains of *P. vivax*. Mathematical models can provide key insights into how treatment strategies may affect *P. vivax* transmission. For example, Robinson et al[10]. highlighted that a mass drug administration (MDA) programme with primaquine and G6PD testing may cause substantial reductions in transmission, but that a mass screen and treat (MSAT) programme with parasitological screening (with either RDTs or PCR) is unlikely to be effective because of the inability to detect hypnozoites. Substantial questions remain about the potential impact of increasing access to anti-hypnozoite drugs for first-line treatment of all symptomatic cases, or if adding MDA rounds to our current set of control tools can help accelerate towards elimination. Assessing the impact of these intervention strategies on population-level transmission will require large, well-designed intervention trials or detailed observational studies. In the interim when such data are not available, mathematical models will play a key role allowing estimation of the effect size of interventions and aiding the design of future population-level intervention studies.

Here, we build on existing theoretical methods for the contribution of relapses to *P. vivax* transmission[24–29], and develop a new, detailed individual-based simulation model of *P. vivax* transmission calibrated to data from many epidemiological studies from PNG and the Solomon Islands. Using this simulation model we investigate the impact of current and future malaria control interventions on *P. vivax*, and the potential impact of future treatment strategies with primaquine or tafenoquine.

## Results

**Model calibration to epidemiological data**. The model captures the key features of the epidemiology of *P. vivax* observed in multiple cross-sectional and longitudinal studies (Table 1), most notably the distinctive peaks in prevalence and clinical incidence in children younger than 10 years of age (Fig. 1). Although the model accurately captures the $PvPR_{LM}$ peak, the $PvPR_{PCR}$ peak is not fully captured. The data and model demonstrate a consistent ordering in the timing of these peaks, with clinical incidence peaking in the youngest age groups, followed by $PvPR_{LM}$, and then by $PvPR_{PCR}$. In higher transmission settings, there is a notable shift of the peaks to younger age groups.

**Model validation**. The results presented in Fig. 1 demonstrate that the model captures some key features of *P. vivax* epidemiology in PNG. However, as the model was explicitly calibrated to these datasets, it may be susceptible to a degree of overfitting. To demonstrate applicability of the model beyond PNG, we compared the model-predicted relationships between *P. vivax* prevalence, entomological inoculation rate (EIR), clinical incidence and hypnozoite prevalence to data from systematic reviews from *P. vivax* studies from across the world with varying age ranges[30,31] (Fig. 2). Although there are standardised methods for measuring *P. vivax* prevalence, estimates of clinical incidence can be very variable according to whether surveillance is undertaken by passive case detection (PCD) or active case detection (ACD) and the frequency of ACD. In particular, more intense ACD will yield higher estimates of *P. vivax* incidence in the same setting[30].

**Table 1 Data from cross-sectional and longitudinal studies used for model calibration**

| Location | Study period | Age (years) | PCR | LM | Clinical | Reference |
|---|---|---|---|---|---|---|
| Cross-sectional data | | | | | | |
| Ngella, Solomon Islands | 2012 | 18 (0.5, 100) | 468/3501 | 127/3501 | 15/3501 | Waltmann[61] |
| PNG; >1500 m | 2000/02 | 16 (0.4, 77) | | 32/664 | 5/664 | Senn[62] |
| PNG; 1000–1500 m | | 17 (0.6, 95) | | 217/2835 | 35/2835 | Senn[62] |
| PNG; 500–1000 m | | 19 (0.0, 87) | | 446/9030 | 93/9030 | Senn[62] |
| PNG; 0–500 m | | 22 (0.1, 99) | | 290/9943 | 109/9943 | Senn[62] |
| Wosera, East Sepik | 1991/92, 1998/99, 2001/03 | 17 (0.1, 80) | 901/2527 | 368/2527 | 24/2527 | Mueller[63] |
| Wosera, East Sepik | 1991/92 | 17.4 (0.1, 87) | | 1207/6782 | | Genton[64] |
| Wosera, East Sepik | 2001/03 | 17 (0.1, 99) | | 1639/15737 | | Kasehagen[65] |
| Madang | 2006 | 14.2 (0.0, 72) | | 204/1227 | 22/1227 | Koepfli[66] |
| Ilaita & Sunuhu | 2006 | 1.7 (0.8, 3.2) | 1433/2129 | 1092/2129 | 133/2129 | Lin[67] |
| Longitudinal data | | | | | | |
| Mugil, Madang | 2004 | 9.3 (4.8, 14.4) | 192/204 | 139/204 | 10/204 | Michon[68] |
| Albinama, East Sepik (placebo arm) | 2008/09 | 7.6 (4.8, 10.4) | 179/257 | 132/257 | 22/257 | Robinson[10] |
| Albinama, East Sepik (primaquine arm) | 2008/09 | 7.5 (4.9, 10.4) | 69/247 | 45/247 | 9/247 | Robinson[10] |
| Wosera, East Sepik | 1998/99 | 16 (0.1, 85) | | 686/1689 | | Kasehagen[69] |

Age is presented as median with range. Samples were tested for parasitaemia by PCR or light microscopy, or for a clinical case of *P. vivax* if accompanied by fever in the last 48 h
*n/N* denotes *n* positive out of *N* samples

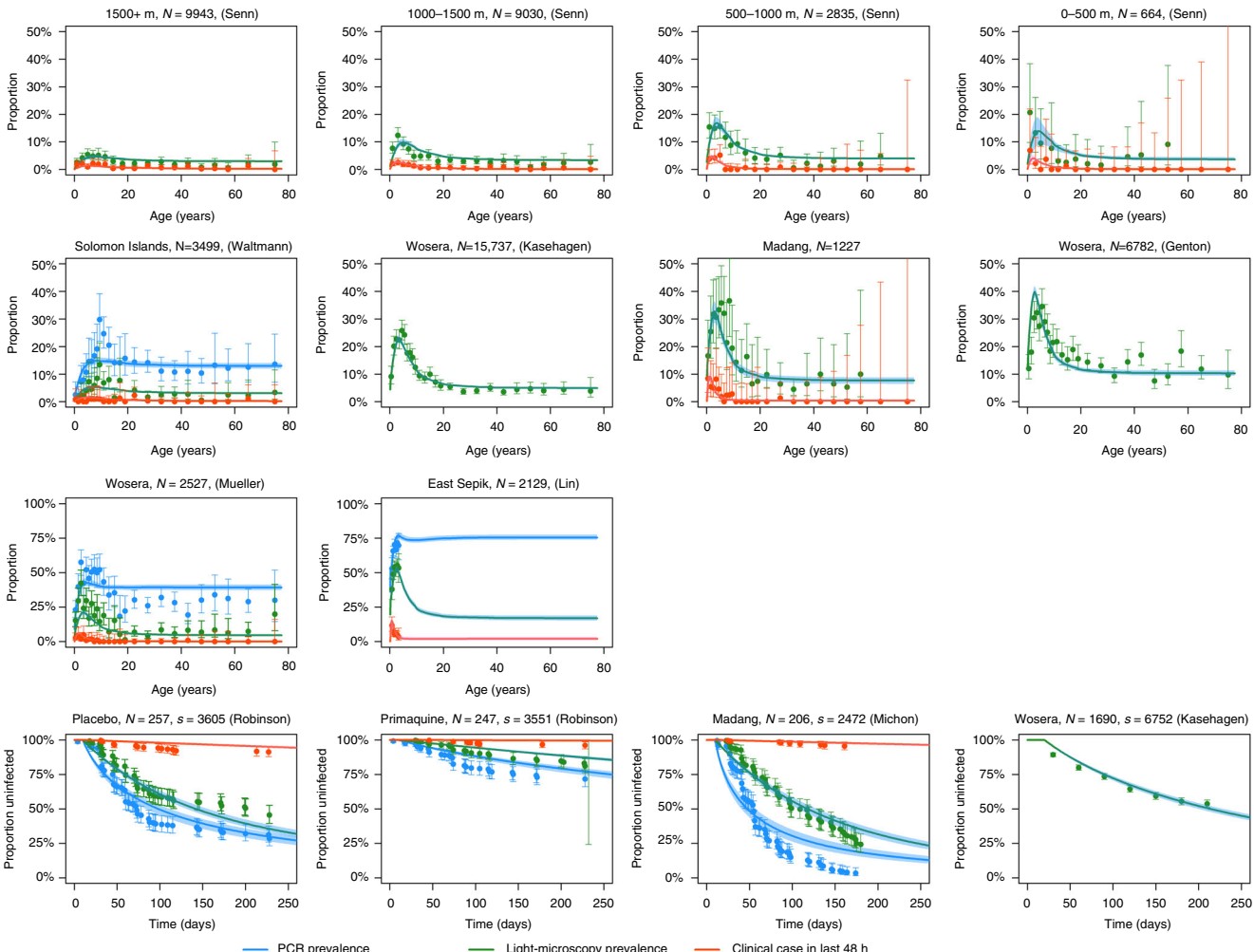

**Fig. 1** Calibration of model to cross-sectional and longitudinal surveys from PNG and the Solomon Islands. For the cross-sectional surveys, the data are presented as age-stratified estimates of prevalence with 95% confidence intervals. The same definition of clinical malaria was used in all cross-sectional surveys: high density parasitaemia and fever (≥38 °C) in the last 48 h. The number of individuals in each cross-section is denoted *N*. For the longitudinal surveys in the bottom row, the data are presented as Kaplan–Meier estimates of proportion infected with 95% confidence intervals. The number of individuals included in longitudinal follow-up is denoted *N* and the total number of samples denoted *s*. The solid curves show the posterior median model prediction, and the shaded regions denote the 95% credible intervals

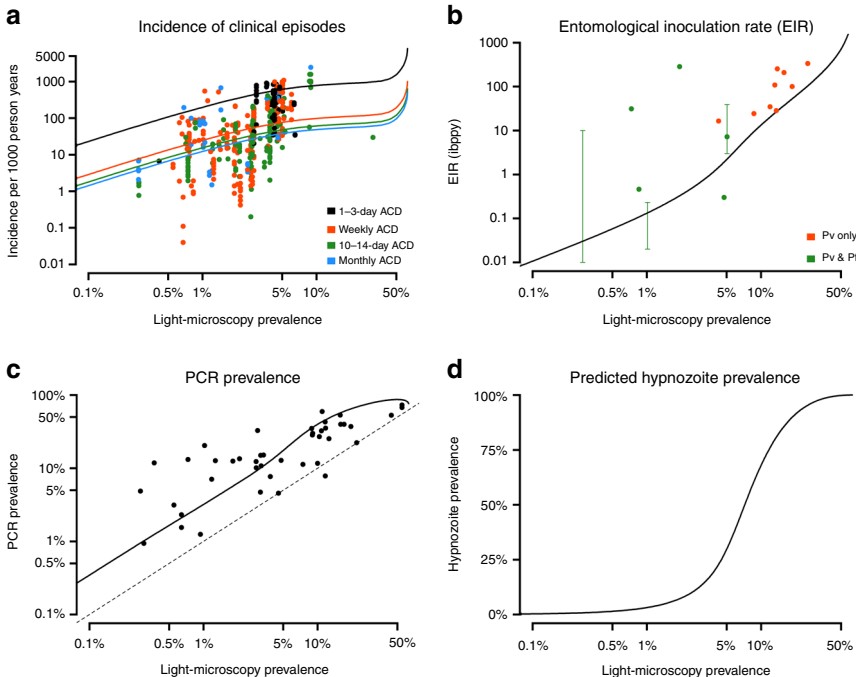

**Fig. 2** Validation of model to data from systematic reviews. Data are shown as points and model predictions as solid lines. **a** Relationship between $PvPR_{LM}$ and clinical incidence based on data reviewed by Battle et al.[30]. Studies included in this review came from throughout the world, had various age ranges, and differences in the frequency of active case detection (ACD) for clinical cases of *P. vivax*. **b** Relationship between EIR and $PvPR_{LM}$ compared to data reviewed by Battle et al. plus data from a study in Papua New Guinea by Burkot et al.[60]. Orange points denote studies where infectious mosquitoes were confirmed as *P. vivax* positive, and green points denote studies where mosquitoes were identified as infectious but without *Plasmodium* species identification. In some studies, a range was provided instead of a point estimate. **c** Relationship between $PvPR_{LM}$ and $PvPR_{PCR}$ compared to data reviewed by Moreira et al.[31]. Data are from throughout the world and are based on studies with various age ranges. **d** Model-predicted relationship between $PvPR_{LM}$ and hypnozoite prevalence. As it is currently not possible to directly detect hypnozoites, this relationship cannot be formally compared to any data

**Simulation of intervention scenarios**. Figure 3 shows past and future projections of *P. vivax* in the provinces of PNG. The black curves denote estimated $PvPR_{LM}$ based on provincial level data from past LLIN distribution campaigns, under a scenario where nets are not replaced (50% of nets are still in use after 19.5 months). In all situations, if LLIN campaigns do not continue, we predict there to be substantial rebounds in *P. vivax*—exceeding even pre-intervention baselines. This rebound is attributable to waning levels of immunity and the lack of acquisition of immunity in young children born since 2009 after the reduction in *P. vivax* transmission. The stochastic uncertainty of this rebound is shown in Supplementary Figures 12 and 13.

A range of future intervention scenarios are simulated. In a situation where LLINs are distributed every 3 years at 50% coverage, we predict that transmission can be reduced to <0.1% $PvPR_{LM}$ in low transmission provinces. In provinces with moderate transmission, we predict that transmission will remain suppressed but not interrupted. In high transmission provinces we predict that prevalence will slowly rebound over time. Increasing coverage levels to 80% is predicted to prevent this rebound. Vector control is predicted to be effective across all of PNG, however impact is expected to be more limited in provinces with a high proportion of outdoor and early biting *Anopheles farauti* s. s. (predominantly coastal and island provinces such as New Britain and New Ireland). The introduction of the 8-AQs primaquine or tafenoquine into first-line treatment regimens for symptomatic *P. vivax* cases was simulated from 2020. This was projected to cause further reductions in $PvPR_{LM}$ with the greatest reductions expected in provinces where transmission remains high even after the introduction of vector control.

As the assumptions regarding vector control are critical to the model predictions, a number of sensitivity analyses were

performed. Assuming slower loss of LLIN adherence[32,33] resulted in marginally greater predicted reductions in *P. vivax* transmission (Supplementary Figures 9 and 10). A counterfactual scenario was considered where LLINs were assumed not to be effective, with the reductions in $PvPR_{LM}$ instead being attributable to a massive scale up in primaquine treatment with screening for G6PD deficiency from 2011 (Supplementary Figure 11). In such a scenario, predicted reductions in $PvPR_{LM}$ were not consistent with the magnitude of the observed reductions, suggesting that the increases in LLIN coverage are attributable for the reductions in *P. vivax*, in agreement with epidemiological studies[13,14].

**Provincial stratification in Papua New Guinea**. Figure 4a, b shows $PvPR_{LM}$ across PNG in 2010 and 2014 based on household prevalence surveys. Figure 4c shows projected $PvPR_{LM}$ in 2025 if LLIN distribution campaigns at 80% coverage are maintained every 3 years. Notably, we predict that in a large proportion of mainland PNG, *P. vivax* will be suppressed to transmission levels less than 0.1% $PvPR_{LM}$. The most challenging areas consist of the islands of New Britain, New Ireland and Bougainville, as well as the coastal provinces of Sandaun and Milne Bay. All of these provinces had high baseline levels of transmission as well as a high proportion of *An. farauti* s. s. Fig. 4d shows the combination of interventions required to reduce $PvPR_{LM}$ to less than 0.1% by 2025. It is assumed that LLIN coverage is first scaled up, followed by scaling up access to tafenoquine with screening for G6PD deficiency in first-line treatment regimens from 2020, and finally a tafenoquine MDA with screening for G6PD deficiency at 80% coverage in 2020. Detailed output of the simulations are provided in Supplementary Figures 12–20. Although combinations of interventions were projected to be sufficient to obtain

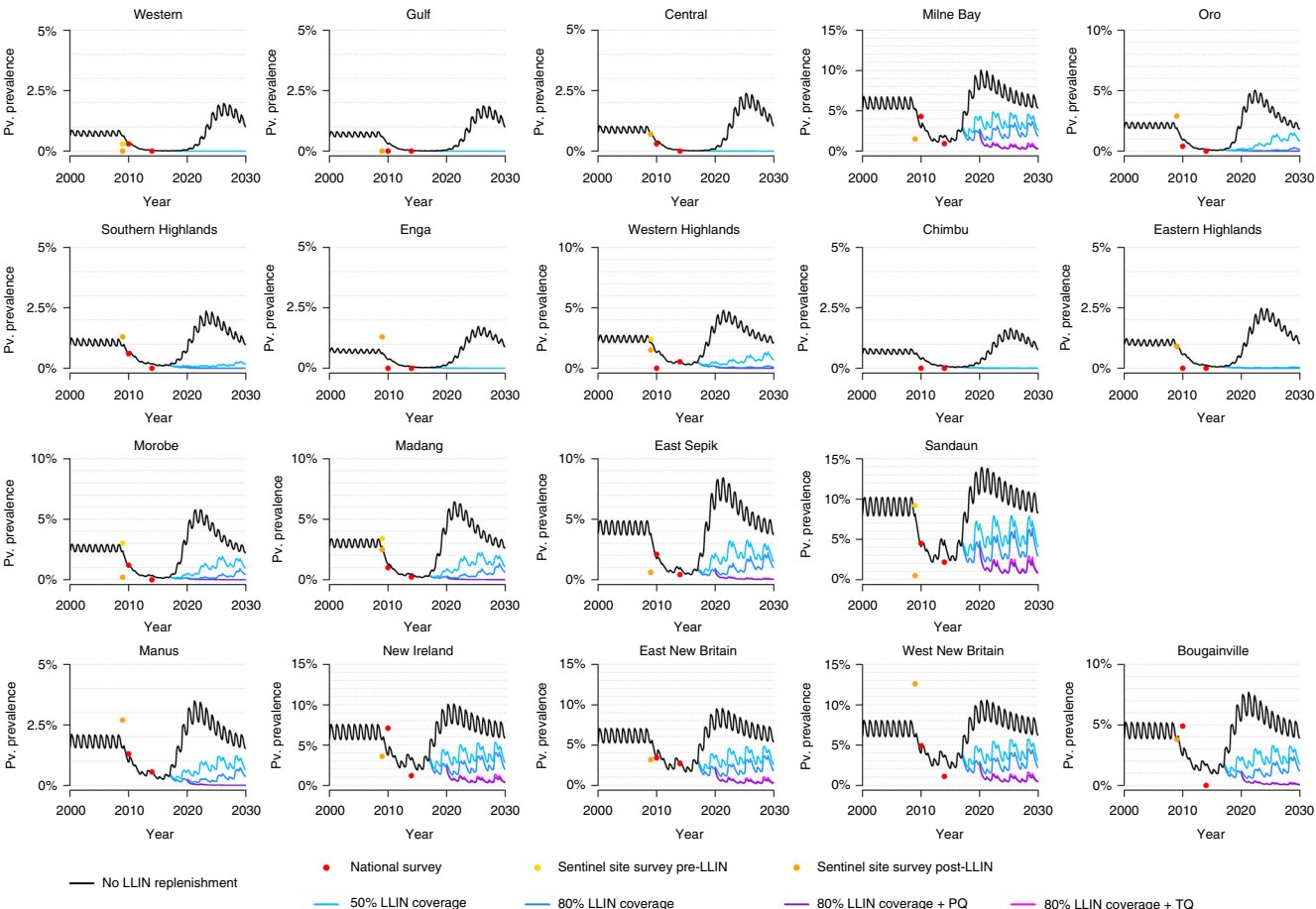

**Fig. 3** Predicted $PvPR_{LM}$ in Papua New Guinean provinces using individual-based model. Model predictions are based on the median of 100 stochastic simulations. Data are from household prevalence surveys in randomly selected villages[14], and surveys from a number of sentinel villages either before or after LLIN distribution[13]. The black curves denote the model-predicted scenario if LLINs are not replaced. In the LLIN campaigns, nets are assumed to be distributed every 3 years, with 50% of nets still in use after 19.5 months. Primaquine (PQ) or tafenoquine (TQ) with accompanying G6PD screening are assumed to be include in first-line treatment regimens from 2020, with 50% of individuals experiencing a clinical episode of *P. vivax* being tested and treated

$PvPR_{LM} < 0.1\%$ in most provinces, there are several provinces where we do not expect reduction in transmission to this level (Fig. 4d).

**Controlling *P. vivax* in high transmission settings**. PNG contains areas with the most intense *P. vivax* transmission in the world[5,6], which may pose some of the greatest challenges to malaria elimination worldwide. Figure 5 presents model projections of *P. vivax* under a range of intervention scenarios in just such a high transmission setting in New Ireland—an island province with a high proportion of early biting *An. farauti* s. s.[34]. Large reductions in *P. vivax* transmission are expected to be achievable by scaling up LLIN coverage and access to tafenoquine treatment, but we do not expect these interventions to be sufficient to interrupt transmission. Tafenoquine, administered through either first-line treatment or MDA campaigns, is projected to cause a 58–86% reduction in cases of *P. vivax* (Fig. 5e). Notably, although MDA campaigns produce sharp reductions in transmission they necessitate a large degree of over-treatment (Fig. 5f), i.e. drugs given to individuals without any parasites.

**Discussion**
PNG is at a key stage in its journey towards controlling and eliminating malaria, having achieved large reductions in *P.*

*falciparum* and *P. vivax* prevalence throughout the country following mass distribution of LLINs[14]. Despite this encouraging progress, there are real risks of widespread rebounds, particularly for *P. vivax*. Notably, following increased coverage of LLINs there were large reductions, but not elimination, of both *P. vivax* and *P. falciparum* in villages in Madang province, followed by a rebound in *P. vivax*[35] with more recent evidence suggesting a slower rebound for *P. falciparum*. The reasons for this rebound are unclear, but may be due to user fatigue with LLINs, stock-outs of anti-malarial drugs, a shift of mosquito biting to early evening[36] or waning population-level immunity. A key result of this analysis is that if LLIN replacement does not continue, then *P. vivax* rebounds will occur leading to higher levels of transmission than before LLIN distribution. Those most at risk will be young children born since 2009 who have grown up in times of low malaria transmission. Rebounds have been repeatedly observed in over 60 countries, including the PNG highlands[37,38] usually following reductions in coverage of malaria control interventions. It is therefore urgent that funding for continued nationwide LLIN distribution be maintained.

Our second key result demonstrates that although maintaining high levels of LLIN coverage will prevent rebounds, it will not be sufficient to eliminate *P. vivax* transmission in many provinces, particularly areas with a large proportion of *An. farauti* s. s. Given the unpalatable options of potentially catastrophic rebounds or

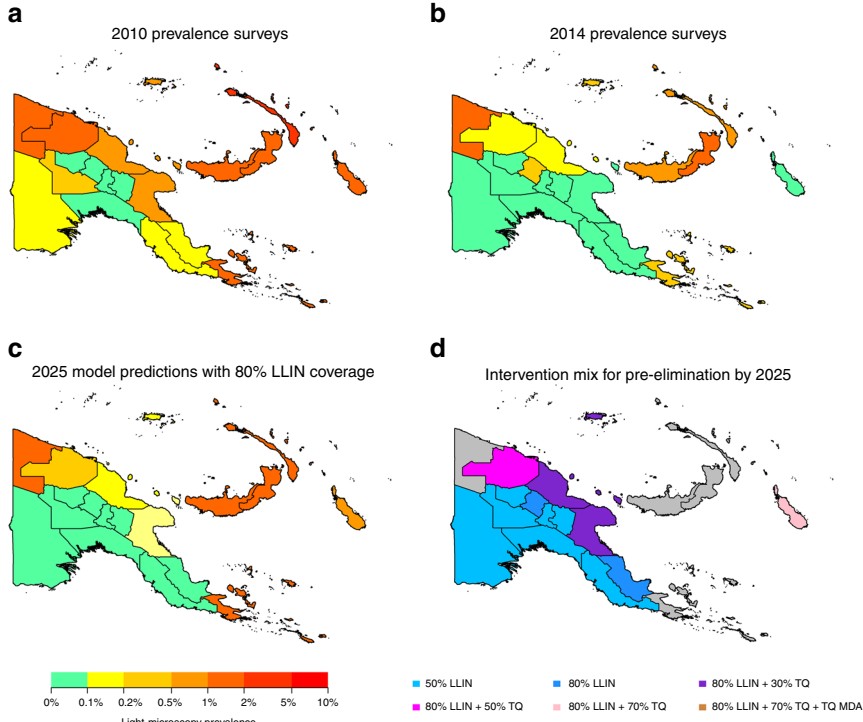

**Fig. 4** Predicted impact of combinations of interventions in Papua New Guinea. Model predictions are based on the median of 100 stochastic simulations. **a, b** Estimated $PvPR_{LM}$ in Papua New Guinea provinces based on household surveys from 2010 and 2014. **c** Model-predicted prevalence in 2025 under a scenario where LLINs are distributed every 3 years at 80% coverage levels. **d** Combinations of interventions required to obtain pre-elimination (defined as prevalence < 0.1%) by 2025. Grey shading indicates that the interventions considered were predicted not to be sufficient to reduce prevalence to <0.1%

indefinite, expensive vector control, there is a strong case to consider combinations of LLINs with new and existing interventions. First-line treatment of symptomatic malaria cases through health centres and aid posts has long been a crucial platform for delivering interventions in PNG[39]. Although primaquine is available in some health facilities, there is little available data on its routine use. The simulations presented here demonstrate that increasing primaquine availability can cause substantial reductions in transmission. However, these high prevalence settings are not representative of the many settings in these regions with low or epidemic *P. vivax* transmission, where it will be important to identify the minimum level of malaria control coverage interventions that can prevent transmission.

Two of the major barriers to effective primaquine treatment are problems with adherence and dosing, and the failure to clear hypnozoites in individuals with a low CYP2D6 metabolizer phenotype[22]. Tafenoquine is not subject to these limitations, although its single high dose means that it could not be provided without G6PD screening. Phase 3 trials of tafenoquine have recently been completed which will provide key data on efficacy in treated individuals, however there is an important need to consider the impact on population-level *P. vivax* transmission. Two key strategies are to increase availability of tafenoquine treatment and G6PD testing in health facilities for first-line treatment of symptomatic episodes, or to presumptively treat entire populations with MDA programmes. The expected effect size will depend on several factors including transmission intensity and the presence of other interventions. In the high transmission setting modelled in Fig. 5, we predict that tafenoquine with screening for G6PD deficiency administered as part of first-line treatment or through MDA could cause a 58–86% reduction in *P. vivax* cases. An important limitation of MDA programmes with 8-AQs is that many people unlikely to have hypnozoites end up receiving a potentially dangerous treatment

regimen. Such a degree of over-treatment has been acceptable in past successful *P. vivax* elimination programmes based on MDA with primaquine in China and former Soviet countries[40,41] where detailed population-level surveillance for side effects was possible. However comparable campaigns have not taken place in resource poor settings with intense transmission of tropical strains of *P. vivax*.

Controlling and eliminating malaria requires addressing all species, including *P. ovale* and *P. malariae*[42]. Although *P. falciparum* has long been seen as the primary contributor to the public health burden of malaria, there is increasing evidence that *P. vivax* is responsible for a comparable degree of mortality, primarily due to high levels of anaemia in patients suffering from multiple relapses[43]. Although a malaria control programme designed with the aim of reducing *P. falciparum* transmission will also succeed in reducing transmission of other *Plasmodium* species, the fact that *P. vivax* begins to dominate as transmission is reduced means that there is an important need to incorporate *P. vivax* interventions aimed at tackling the hypnozoite reservoir.

There are a number of limitations to the model and analysis presented here. Most notable is the choice of province as the spatial transmission unit in PNG. These provinces are geographically large with sizable populations and considerable diversity in malaria transmission. Highland provinces have very variable transmission where pockets of high transmission persist in valleys with lower altitude. Although we have used the best epidemiological data currently available, the future strengthening of health information systems in PNG will allow for a more refined approach with simulation of Districts or Local Level Government areas[44]. Although a province as a whole may have very low numbers of *P. vivax* cases, there may be sub-regions with ongoing transmission. An improved understanding of spatial heterogeneity would greatly aid targeted elimination efforts. Inter-provincial travel is also likely to provide a barrier to

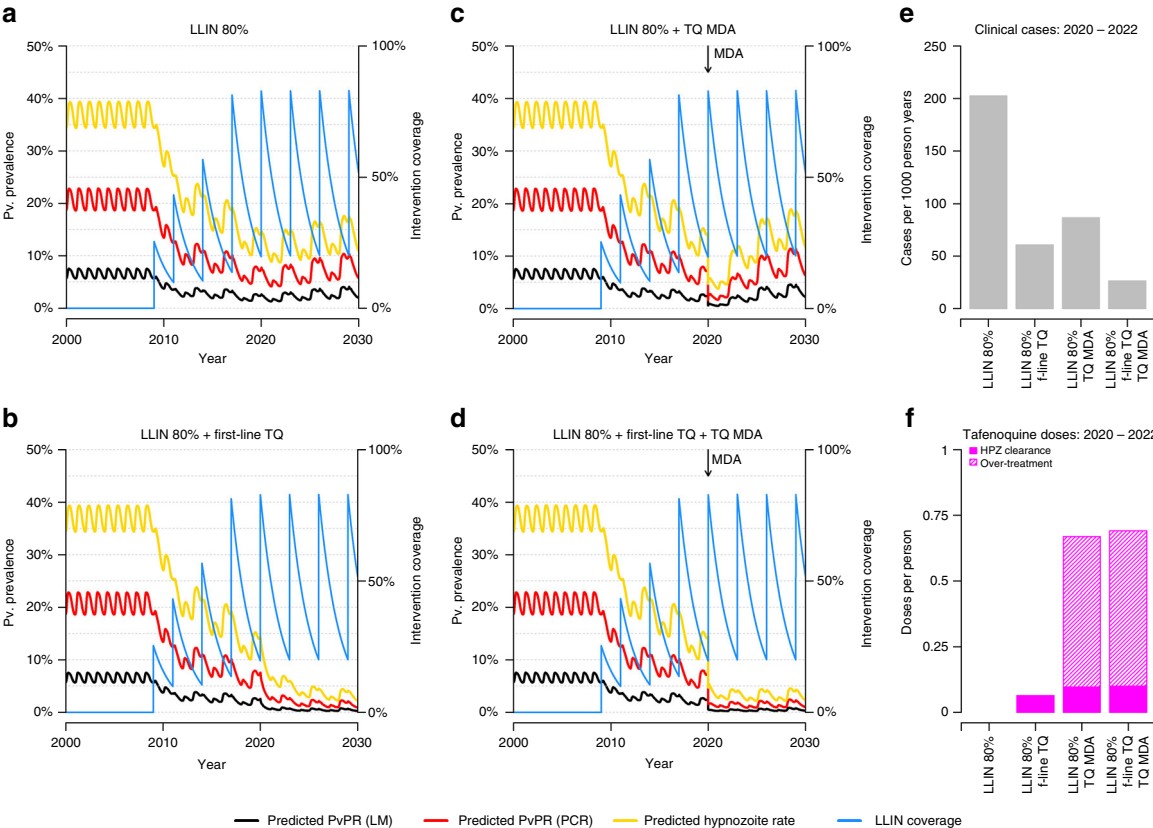

**Fig. 5** Individual-based model projections of *P. vivax* transmission in New Ireland under a range of intervention scenarios. Model predictions are based on the median of 100 stochastic simulations. **a** LLIN campaign every 3 years at 80% coverage. **b** LLIN campaigns plus introduction of tafenoquine (TQ) into first-line treatment regimen from 2020 such that 50% of clinical cases are treated. **c** LLIN campaigns plus mass drug administration (MDA) with tafenoquine at 80% coverage in 2020. **d** LLIN campaigns with TQ incorporated in first-line treatment regimen and TQ MDA. **e** Estimated clinical cases per 1000 person years for the period 2020–2022 under the intervention scenarios considered. **f** Estimated number of TQ doses per person over the period 2020–2022

elimination efforts, although additional simulations accounting for travel did not produce substantially different estimates of the impact of interventions in each province. The accuracy of predictions is further limited by the data available for model calibration. Although there is very rich data on the epidemiology of relapses and age-stratified prevalence and clinical incidence, there are some key areas of uncertainty in our understanding of *P. vivax* transmission. Chief amongst these is understanding the contribution of low-density blood-stage infections to onwards transmission to mosquitoes[45], systematic differences in blood-stage infections resulting from mosquito bites or relapses, and the duration of naturally acquired immunity. Furthermore, we do not account for the genetic diversity of *P. vivax* parasites, which typically exhibits greater diversity than *P. falciparum* parasites in co-endemic regions, suggesting that in comparison to *P. falciparum*, transmission of *P. vivax* is stable with a large effective population size[46,47].

The representation of mosquitoes in the model is by necessity a simplified one. The three primary malaria vectors *An. farauti* s. s., *An. punctulatus* and *An. koliensis* are accounted for, however there are many other species that likely play important roles in sustaining transmission. For the three primary species, there was detailed data from PNG and the Solomon Islands on key entomological parameters such as the human blood index, biting times and seasonality (reviewed in detail in Supplementary Table 3). A key limitation is the absence of data from these regions on mosquito interactions with LLINs, comparable to measurements from experimental hut trials for African malaria

vectors[48]. More generally, a key challenge for vector control in *P. vivax* endemic settings is the development of strategies for targeting outdoor biting mosquitoes. This need may be met in the future through the rapid adaptation of new interventions to target outdoor biting being developed in Africa for species such as *An. arabiensis*[49].

The diversity of malaria transmission settings in PNG, ranging from intense transmission in coastal areas to occasional epidemics in the Highlands, makes it an ideal case study for understanding how combinations of malaria control interventions can be used to control and eliminate *P. vivax*. Mathematical models of the transmission cycle accounting for *P. vivax* relapses provide an invaluable tool for exploring the potential impact of existing and future intervention scenarios. Although the model was calibrated against PNG data, its findings are still applicable to other *P. vivax* endemic regions such as South East Asia and South America after adjustments to account for local epidemiological data. Further development and application of these models will aid in the design of malaria control strategies with new interventions, and more critically outline what is achievable with currently available interventions.

## Methods
**Mathematical model of *P. vivax* transmission**. Ross-Macdonald mathematical models of vector-borne disease transmission have previously been extended to incorporate the relapses characteristic of *P. vivax*[24,25]. Here we build on these methods to develop an individual-based simulation model of the transmission dynamics of *P. vivax* using a comparable framework to existing models of *P. falciparum* transmission[50,51]. This framework accounts for heterogeneity and

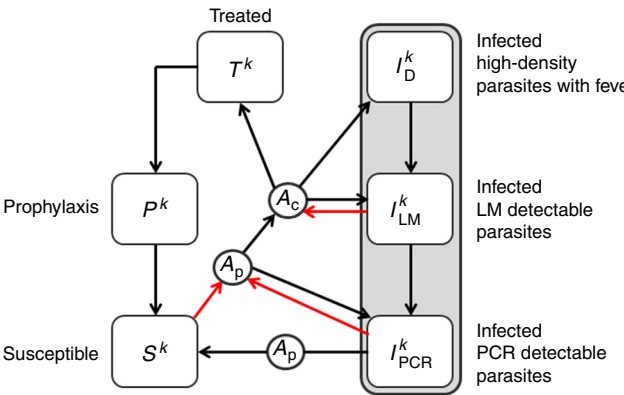

**Fig. 6** Compartmental representation of *P. vivax* transmission model in humans. Infected individuals can be in one of three compartments depending on whether blood-stage parasitaemia is detectable by PCR ($I_{PCR}$), light microscopy ($I_{LM}$) or has high density with accompanying fever ($I_D$). A proportion of individuals that progress to a symptomatic episode of *P. vivax* will undergo treatment with a blood-stage drug (*T*) leading to clearance of blood-stage parasitaemia and a period of prophylactic protection (*P*) before returning to the susceptible state (*S*). The superscript *k* denotes the number of batches of relapse causing hypnozoites in the liver. Red arrows denote new blood-stage infections arising from either new mosquito bites or relapses. Each square denotes a compartment and the circles denote the dependence of transition rates between compartments on levels of anti-parasite immunity ($A_p$) and levels of clinical immunity ($A_c$)

seasonality of exposure to mosquito bites; detailed mosquito bionomics and modelling of larval population densities; demographic age structure; exposure and age-dependent acquisition of immunity against blood-stage infection and clinical episodes; maternally-acquired immunity; and treatment of clinical cases and drug prophylaxis. A key advantage of using an individual-based framework is the ability to account for the complexity that arises when individuals have different combinations of interventions, levels of immunity and genotypes[52]. Figure 6 shows a compartmental representation of the model. Notably the infected state is ordered according to whether an individual's blood-stage infection is detectable by PCR ($I_{PCR}$), detectable by light microscopy ($I_{LM}$) or whether they are undergoing a clinical *P. vivax* episode ($I_D$).

Relapses are described by two key epidemiological parameters: the relapse rate (*f*) and the liver-clearance rate ($\gamma_L$). It is assumed that following an infectious mosquito bite, the liver will remain infected with hypnozoites for an expected $1/\gamma_L$ days, with a relapse occuring on average every $1/f$ days[24]. We allow for super-infection with batches of hypnozoites originating from different mosquito bites. If an individual has *k* batches of hypnozoites then relapses will occur at rate *kf* and the number of batches will reduce from *k* to *k* − 1 at rate $k\gamma_L$ until the liver has been cleared of infection with hypnozoites.

Individuals acquire immunity at a rate proportional to the force of blood-stage infection (with contributions from both new mosquito infections and relapses). Two forms of immunity are accounted for: (i) anti-parasite immunity which reduces the probability that a blood-stage infection will achieve sufficiently high densities to become detectable by light microscopy and increases the rate of clearance of blood-stage infection ($A_p$ in Fig. 6); and (ii) clinical immunity which reduces the probability that a blood-stage infection will progress to cause a case of clinical malaria ($A_c$). Complete mathematical details of the model are provided in the Supplementary Information.

The three most important malaria vectors in PNG are *An. farauti* s., *An. koliensis* and *An. punctulatus*[53]. The population dynamics of these species are modelled assuming density-dependent competition within seasonally varying larval breeding sites[54]. The feeding, resting and oviposition behaviour of adult female mosquitoes are described by a number of key entomological parameters including life expectancy, human blood index (HBI), proportion of day time biting, and indoor resting preferences. See Supplementary Table 3 for full details.

**Model fitting**. The model was fitted to age-stratified data on *P. vivax* prevalence by PCR or light microscopy and clinical cases of *P. vivax* from 56,772 individuals from 11 cross-sectional or longitudinal studies from across PNG and the Solomon Islands (Table 1). The parameters describing the transitions between human states and the acquisition of immunity were estimated by fitting the equilibrium solution of the deterministic model to the data using Bayesian Markov Chain Monte Carlo

(MCMC) methods. All prior and posterior parameter estimates are provided in Supplementary Table 2.

The impact of interventions was simulated using the stochastic, individual-based implementation of the model with a population size of 100,000, initialised using the equilibrium solution of the deterministic model. For every intervention scenario considered, the model was run 100 times with results presented as the median and 95% range of these simulations. The large population size resulted in a limited range of stochastic variation[55].

**Intervention models: treatment**. The existing malaria treatment guidelines in PNG recommend that malaria cases confirmed positive for *P. vivax* or *P. ovale* be administered primaquine at 0.25 mg/kg for 14 days. Primaquine is sporadically available across the country, there is no access to G6PD deficiency testing, and levels of adherence are unknown[11]. The incorporation of the 8-AQs primaquine and tafenoquine into first-line treatment regimens was simulated in the model. The full treatment pathways are provided in Supplementary Figure 5 and summarised as follows. It was assumed that a 14-day primaquine treatment regimen has 70% effectiveness at clearing hypnozoites (due to poor adherence), and that a single dose of tafenoquine has 100% effectiveness against hypnozoites. Furthermore, tafenoquine is assumed to provide 60 days of prophylaxis against further blood-stage or liver-stage infections[23]. We assumed that 8-AQs were not given to pregnant women (about 4% of the population) or children under 6 months of age (about 2.2% of the population). We further assumed that testing for G6PD deficiency was available, e.g. through a test similar to CareStart[20]. The G6PD allele frequency in PNG was assumed to be $q_{G6PD} = 7.4\%$[18]. Therefore $0.5 \ast q_{G6PD} = 3.7\%$ of the population are G6PD deficient males, $0.5 \ast q^2_{G6PD} = 0.27\%$ are G6PD homozygous deficient females, and $0.5 \ast 2 \ast q_{G6PD} \ast (1 − q_{G6PD}) = 6.9\%$ are G6PD heterozygous deficient females. We assumed that 5% of the population had a low CYP2D6 metabolizer phenotype, where primaquine was not efficacious, but that tafenoquine retained its efficacy[56]. Assuming the availability of treatment and G6PD testing, this leads to an effectiveness at clearing hypnozoites of 55.3% for primaquine and 83.2% for tafenoquine.

**Intervention models: vector control**. An existing model of the effects of LLINs on the behaviour of *Anopheles* mosquitoes was incorporated into the model[50,57]. We assume that LLINs have three effects on adult mosquitoes: (i) killing mosquitoes that land on nets; (ii) repelling and possibly diverting mosquitoes to an animal blood host due to either insecticide irritation or the physical barrier of the net; and (iii) lengthening the duration of the gonotrophic cycle leading to a reduced oviposition rate. This model was parameterised to reflect the behaviour of the three primary malaria vectors in PNG: *An. farauti* s. s., *An. koliensis* and *An. punctulatus*[53]. The metric of LLIN coverage employed here is based on use: the proportion of individuals who slept under an LLIN last night[58]. A critical parameter for the effectiveness of LLIN campaigns is the duration of LLIN usage. We assumed that loss of adherence to LLINs occurs at a constant rate so that 50% are still being used after 19.5 months based on data on the age distribution of nets collected in PNG[59]. This duration is shorter than in African data where it was estimated that 50% of LLINs are lost after 23 (20–28) months[32], and also shorter than older recommended assumptions of 36 months from the Roll Back Malaria Harmonization Working Group[33]. Additional sensitivity analyses with these longer durations were simulated.

**Application to PNG**. In 2004 PNG received a grant from The Global Fund to Fight AIDS, Tuberculosis and Malaria to initiate country-wide free distribution of LLINs. This led to large increases in LLIN ownership with 64.6% of households owning a net by 2009[12]. In our study we define LLIN coverage according to LLIN use—the proportion of individuals surveyed who slept under a LLIN during the previous night. Data on coverage are informed by use surveys[12], and distribution data from Rotarians Against Malaria—the organisation responsible for nationwide distribution campaigns. The key metric of *P. vivax* transmission intensity is prevalence by light microscopy ($PvPR_{LM}$). Transmission intensity in each province was measured via household prevalence surveys in at least two randomly selected villages from each province[6]. These surveys were supplemented by measurements from a selection of sentinel sites before LLIN distribution and surveys taken shortly after LLIN distribution[13]. Maps of PNG provinces were generated in R using shape files from http://otlet.sims.berkeley.edu/imls/world/PNG/. In each province *P. vivax* transmission was simulated using the individual-based model. To explore the potential role of inter-provincial travel, we performed a sensitivity analysis where all provinces were simulated simultaneously, accounting for travel of people between provinces by air, road and sea (see Supplementary Information section on inter-provincial travel in PNG).

**Data availability**. The C++ code for implementation of the individual-based model, the model for inter-provincial travel and the data used for model calibration is available for download from GitHub @MWhite-InstitutPasteur (https://github.com/MWhite-InstitutPasteur/Pvivax_IBM).

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

## Acknowledgements

This work was supported by a Population Health Scientist fellowship awarded to M.T.W. from the MRC (MR/L012170/1), and the NIH as part of the Asia Pacific International Centers of Excellence in Malaria Research. This work was informed by a series of meetings between Imperial College London, the Papua New Guinea National Malaria Control Programme and other key stakeholders, supported by The Global Fund to Fight AIDS, Tuberculosis and Malaria.

## Author contributions

Performed the analysis: M.T.W. Designed the analysis: M.T.W., P.W., S.K., A.G., I.M. Data collection M.W.H., T.F., A.W., M.L., L.J.R., I.M. All authors contributed to writing the manuscript.

## Additional information

**Competing interests:** M.T.W., P.W. and A.G. received a consultancy payment from The Global Fund for coordinating a workshop in Port Moresby to provide advice on mathematical modelling to the Papua New Guinea National Malaria Control Programme. M.T.W., P.W. and A.G. declare that they have no other competing interests. All remaining authors declare no competing interests.

