## [Peer Review File · Nature Communications]

Reviewers' comments:

Reviewer #1 (Remarks to the Author):

This is an interesting paper, which presents a thoughtful calibration of a well-described model to a large body of high quality data. This work is used as a platform to interpret recent changes in vivax prevalence in PNG and project the impact of different scenarios.

The supplemental appendix is fantastic and sets a very high bar for transparency in modeling work in general.

The calibration of the immune model is quite convincing. My main concern is the confidence with which changes in prevalence between sentinel sites and national surveys can be attributed predominantly to ITN effects. All the future projections would quantitatively differ to the extent that other variables have played a role.

In the Thai/Myanmar border site (L280) the failure to replace LLINs is predicted to lead to rebounds in transmission. Nguitra et al. shows 90.5% net usage in Thailand, but prevalence the same in users vs. non-users. Significant travel/gender/age risk factors for Pf, with recent travel not significant in Pv (but perhaps because of relapse?) Implications for these projections in Thailand?

Minor comments below:

- * Would you expect a seasonality that included more random noise to result in any differences in either the calibration or projection scenarios compared to the smooth seasonal forms shown in the supplement?
- * Mosquito behavioral parameters vary dramatically among entomological sites for any given species. How does this impact how we think of likely ITN effect sizes? Do we think the 3-year lifespan assumptions are accurate in PNG -- they don't appear to be the norm in Africa (Bhatt, eLife).
- * L291: was Pf completely gone (needed to re-establish from outside) whereas vivax rebounded from omnipresent liver-stage reservoir? Are there other examples in history where the Pv/Pf resurgence difference was observed (Cohen 2012?)
- * L299-301: While there is little data on primaquine usage, how much could it have increased/decreased over the 2009-2014 survey period? What impact could be attributed to that?
- * L332: It might be worth citing some of the recent parasite genetics vivax literature? Pava, Jennison, Pearson, etc.
- * While you say there is not enough data to go more granular than province scale, how would the known variation impact these analyses? Is 20k the appropriate "effective

population size" for PNG meta-population dynamics? Especially as it relates to reducing prevalence to $< 0.1\%$ (L235)?

Reviewer #2 (Remarks to the Author):

This is a clear, well-written paper, and the conclusions are supported by the modeling results. The paper has practical deliverables that will be of use to the National Malaria Control Programme of PNG. This work should be published but it belongs more in Nature Scientific Reports than Nature Communications. Using mathematical models, the paper confirms well-known and expected results in malaria epidemiology. The quantitative guidance is of use to the PNG NMCP, but the paper does not present a major advance in malaria epidemiology or mathematical modeling.

1. Abstract - increased bed net coverage is predicted to reduce transmission but rebounds will occur if interventions are removed. This is an expected and well-understood result. The last sentence of the abstract aims to say something about future options for vivax control, but it is very non-specific.
2. The authors should make clear what parts of the individual-based model (IBM) were developed for this paper, and what parts were from the original work (ref 24). Also, it does seem appear that simulations for some of the figures may have been run with a more straightforward ODE model (maybe equations S1) as there is very little stochasticity in Figures 4 and 6. I can't be sure, because the figure captions are missing for the figures in the main text.
3. A major limitation is that the IBM runs are done with 20,000 individuals and replicated 3 times. For a Nature Communications publication, you should have the full PNG population size, some appropriate spatial resolution (province-level is probably fine here), and you should be doing 50 or 100 runs per scenario so that you can look at variation in hard-to-pin-down processes like rebounds. The rebound time will not always be the same in individual-based model runs.
4. It is good that an adherence parameter is included in the model, but a sensitivity analysis should be run here.
5. Model validation paragraph (second paragraph of results): this is excellent. Very important to acknowledge that with many spatial parameters overfitting will occur. Great work introducing this extra out-of-sample validation. This is rarely done with IBMs.
6. Since you have prevalence data for both 2010 and 2014 in PNG, you can also do an extra validation to see if the you get close to the 2014 final condition when you start with 2010 as an initial condition. Do you have enough information on interventions, drug coverage, bed net coverage, etc. from 2010 to 2014?
7. I would remove the Thailand and Brazil simulations in Figure 7. This has the potential to

be misused if picked up by the wrong person, and you haven't done all the background legwork, data assembly, descriptive epidemiology, spatial analysis that you have done for PNG. Figure 7 just shows general country level prevalence trajectories, under five scenarios, with no uncertainty shown.

8. No movement between provinces in PNG. Again, for a Nat Commun paper, you would need to include this, even if it's a sensitivity analysis on a poorly known migration/movement rate.

9. You mention a handful of times that the cohort of children born post-2009 may have low immunity after an intervention, and that they will make up the major part of the rebound if interventions are relaxed. Why not design an intervention specifically for this cohort? This would be a novel and as yet unstudied intervention type.

At Nature Scientific Reports, I would accept this paper with very few revisions.

Reviewer #3 (Remarks to the Author):

The paper discusses a mathematical model of malaria control interventions on *P. vivax*. The paper is largely focused on Papua New Guinea, but makes some suggestions as to overall *P. vivax* interventions globally. Overall more focus on the issues of *P. vivax* is good, and it is an overall interesting paper, but I think suffers from both a glut of details and obfuscation of the important details. The results are not particularly novel even if they are adding in the latest drugs. In addition, I don't think the authors really attempt to examine lots of alternatives to basic assumptions which would help in understanding the efficacy of interventions.

Major Comments:

1. The results are somewhat dispiriting and similar to prior results. The finding that MDA and vector control is not enough to eliminate malaria is not particularly different from past analyses, perhaps it would be useful/instructive to find potential solutions within the model (i.e., under what scenarios can elimination be achieved) and then see if those are possible to implement in real life.

2. This paper I feel is a good demonstration of the problems in publishing today. The paper itself is light on details, while the supplementary materials reads like a thesis. I mean the supplementary material is 40 pages long and around 15,000 words. It would be much better to reduce extraneous components of the paper and somehow reduce the supplementary material. Perhaps by publishing the model as a separate paper in a different journal.

3. Despite the lengthy supplementary text, a major component of the model development is completely glossed over: the model results described in the paper are based on a stochastic individual-based formulation of the model, however the development of this individual-based model is not described anywhere. This should be fully detailed as to the choices made on timing, how the code was translated, how the numbers of individuals were chosen (e.g., why such a small number), and how individuals were allocated between provinces. In addition, a full explanation should be made regarding why the authors chose to use a

stochastic model implementation of a deterministic model. What was the benefit? Why go through this trouble? It is not clear to me why this would be the case. Lastly, for such a complex model, the code for producing the model should be made available as part of the project. Preferably on a code-hosting site such as bitbucket/github. In this day in age, it is not really appropriate to create complex mathematical models without also posting the code.

4. Despite the attention to detail in the model, the major assumptions greatly reduce its utility. The lack of mixing between provinces and the free mixing within provinces seem to be major issues that really hinder this model. Heterogeneous biting and geographic distances from water sources have long been shown to be important in *P. falciparum* and there is no expectation that this would differ for *P. vivax*. It was actually quite disappointing to read how seemingly far behind this model was in terms of those aspects. Realistically, a simple model would have come to nearly the exact same conclusions, so again I am just totally mystified by the decision to make this an individual-based model. I could understand it if that resulted in some novel interpretation or provided additional value regarding how to implement interventions or included multiple things like heterogeneous biting or geographic variation in biting, but really the result seems that it would be the same if a much simpler model was used.

5. "In particular, we do not account for spatial variation in transmission intensity within each province which is known to vary substantially. Ideally we would be able to model transmission at the lower levels of Districts or Local Level Government (LLG) area. However, there is not sufficient data to meaningfully model malaria transmission at any spatial scale lower than the province level. " spatial data is readily available these days, so breeding sites and thus transmission could be estimated. Even if not exact, how do these factors affect the conclusions. One would expect that heterogeneity could either improve or hurt elimination efforts and that it would be good to use mathematical models to estimate similar levels of incidence for different heterogeneities in transmission and see the differential effects of interventions.

6. The Figure Legends are numbered incorrectly

7. One major assumption is that implementation of MDA could even proceed without G6PD testing. This is mentioned in passing, but seems that should be a larger focus of the sensitivity analysis.

8. The paper is based around PG, cut all the extra stuff about Thailand and Brazil from the paper. Though Figure 3 does suggest that the model gets the general gist of the data, it clearly is not capturing the wide variation across the world, and does not provide a ton of confidence that this can explain global variations. It also just makes a sprawling unwieldy paper less focused. I think this paper should just clearly focus on the issues of PG, and then perhaps in the discussion mention note that it could perhaps be applied in other settings if the data were collected to provide that avenue – that is all that is really necessary.

Minor comments

Figure 2: What is the deal with East Septik, why are there no dots in the older ages?

Figure S3: To my eye the line does not fit the data very well. Why not use something like STL to fit the data? Or perhaps something that is less perfectly regular? Since from the data it appears that seasonality is moving around a bit, how would that affect the outcomes,

would it have any affect? Also it is noted that equation S17 has seasonal equation, but it is S18.

Reviewer #1

The supplemental appendix is fantastic and sets a very high bar for transparency in modeling work in general.

Thanks to the reviewer for acknowledging these efforts. In order to further increase transparency we have included additional details of the implementation of the individual-based model, and also uploaded our model code (with detailed comments) to GitHub: @MWhite-InstitutPasteur.

The calibration of the immune model is quite convincing. My main concern is the confidence with which changes in prevalence between sentinel sites and national surveys can be attributed predominantly to ITN effects. All the future projections would quantitatively differ to the extent that other variables have played a role.

In the manuscript, we have cited data from multiple epidemiological studies across PNG coordinated by several of us (led by M Hetzel). These studies found decreases in malaria prevalence (all species) following increased LLIN coverage. These epidemiological observations and our modelling analysis both strongly point towards the reduction in prevalence being caused by LLINs, however this does not guarantee that LLINs played a causal role.

The other plausible explanation for the observed nationwide reduction in prevalence is the changes to first-line treatment policy in 2011 – 2012. It should be noted that: (i) the change in treatment policy occurred in 2011 – 2012, after the observed reductions; (ii) the switch was from chloroquine, a drug that was still efficacious, to a different efficacious drug regimen (ACTs); and (iii) even though primaquine was recommended for samples positive for *P. vivax* or *P. ovale*, it is rarely in stock and there is strong evidence that primaquine administered without directly observed treatment (DOT) has limited efficacy against relapses. Acknowledging these limitations, we simulated an additional counterfactual scenario where LLINs are not included, but instead access to treatment is dramatically improved in 2011 so that 50% of symptomatic cases receive prompt primaquine treatment. The results of these simulations are presented in the Supplementary Information Figure S11 and the implications discussed in the discussion.

In the Thai/Myanmar border site (L280) the failure to replace LLINs is predicted to lead to rebounds in transmission. Nguitragool et al. shows 90.5% net usage in Thailand, but prevalence the same in users vs. non-users. Significant travel/gender/age risk factors for Pf, with recent travel not significant in Pv (but perhaps because of relapse?) Implications for these projections in Thailand?

This point has become moot in light of Reviewer 2 and 3's recommendations to remove the Thai and Brazilian simulations, but it is important nonetheless. The finding from the epidemiological study by Nguitragool *et al* about the lack of an association between bed net usage and prevalence in a setting with high bed net coverage is not surprising. This is for two reasons: (i) in this region of Thailand (and much of the Mekong region) malaria transmission is forest-based and related to occupation with exposure occurring when individuals are away from home and their bed nets; and (ii) much of the protection that is conferred by bed net usage is due to the population-level reduction in transmission rather than individual-level protection.

* Would you expect a seasonality that included more random noise to result in any differences in either the calibration or projection scenarios compared to the smooth seasonal forms shown in the supplement?

Random noise due to seasonal variation will influence factors such as the timing of the peak of transmission, and the ratio of peak to trough mosquito numbers. While variation in these parameters will lead to substantial variation in mosquito numbers, the variation in *P. vivax* prevalence will be less substantial. To investigate this further we calculated the 95% confidence intervals for the seasonal mosquito patterns – see the revised Supplementary Figure S3 and the response to Reviewer 3 below. To further address the reviewer’s concern, we then sampled 10 parameter values from the estimated range for *Anopheles farauti* and simulated the mosquito numbers and the resulting levels of *P. vivax*. The results of these simulations are presented in the Additional Figure 1 below, where we see how substantial seasonal variation in mosquito numbers, leads to much less variation in prevalence over time.

Additional Figure 1: Ten sampled patterns for *Anopheles farauti* seasonality and the resulting patterns for *P. vivax* prevalence by light-microscopy (black), by PCR (red), and hypnozoite prevalence (yellow).

* Mosquito behavioral parameters vary dramatically among entomological sites for any given species. How does this impact how we think of likely ITN effect sizes? Do we think the 3-year lifespan assumptions are accurate in PNG -- they don't appear to be the norm in Africa (Bhatt, eLife).

The reviewer is absolutely correct that mosquito parameters will vary dramatically between sites, even for the same species within PNG (this can be seen in the data reviewed in Table S3). It is likely that these parameters will impact LLIN effect size, but we believe the reviewer has highlighted the most critical parameter: namely the lifespan of LLINs. Our previous estimate of a half-life of 3 years was based on recommended values by the Roll Back Malaria Harmonization Working Group.

As noted by the reviewer, this duration is longer than that of 23 (20-28) months estimated by Bhatt *et al* based on African data. We were further able to obtain data on the age distribution of Papua New Guinean nets (Katusale *et al*), from which we were able to estimate that the half-life of nets in PNG was 19.5 months, shorter than observed in Africa.

All simulations were therefore repeated assuming an LLIN half-life of 19.5 months. As sensitivity analyses, the results of simulations with an LLIN half-life of 23 months and 36 months (3 years) are also presented in Supplementary Figures S9 and S10.

RBM-HWG. 2014. Malaria implementation guidance in support of the preparation of concept notes for the global fund.

Katusele M, Gideon G, Thomsen EK, Siba PM, Hetzel MW, Reimer LJ. Long-lasting insecticidal nets remain efficacious after five years of use in Papua New Guinea. *PNG Med J* 2014; 57(1-4): 86-93

* L291: was Pf completely gone (needed to re-establish from outside) whereas vivax rebounded from omnipresent liver-stage reservoir? Are there other examples in history where the Pv/Pf resurgence difference was observed (Cohen 2012?)

P. falciparum prevalence was reduced but not eliminated. We have adjusted the text to clarify this point. We have also now noted the suggested paper by Cohen *et al* on malaria rebounds.

* L299-301: While there is little data on primaquine usage, how much could it have increased/decreased over the 2009-2014 survey period? What impact could be attributed to that?

While it is known that primaquine is available in some first-line health facilities (Pulford *et al.* ref 46), there is very little available data on how frequently it is actually used for the treatment of symptomatic episodes. The baseline assumption for treatment was that 30% of symptomatic episodes received effective treatment with a blood-stage anti-malarial. We further simulated increased access to primaquine with G6PD testing from 2011 whereby 50% of symptomatic episodes receive effective treatment. The results of these analyses are presented in the Supplementary Figure S11.

* L332: It might be worth citing some of the recent parasite genetics vivax literature? Pava, Jennison, Pearson, etc.

We agree and have now cited some of the recent *P. vivax* genetics literature, noting that it is an additional limitation of our analysis that we do not explicitly account for genetics.

* While you say there is not enough data to go more granular than province scale, how would the known variation impact these analyses? Is 20k the appropriate "effective population size" for PNG meta-population dynamics? Especially as it relates to reducing prevalence to < 0.1% (L235)?

In our individual-based simulations, greater population sizes are associated with less stochastic variation in mean outcomes. This is especially true as infection becomes rarer (e.g. when prevalence < 0.1%). To investigate this we examined the stochasticity in model predictions for populations greater than 20,000 when interventions reduced prevalence <0.1%. We found that a population size of 100,000 was accurately able to simulate prevalences as low as 0.03% – 0.1% (also note that these are LM prevalences and correspond to hypnozoites prevalence of 0.5% - 1%).

We therefore increased the population size to 100,000, and increased the amount of repeat simulations from 3 to 100 to better characterise stochastic variation. The exception to this was in Manus Island province which had a population of 60,485 < 100,000 according to the 2011 census.

Using methods from population genetics, Jennison *et al* were able to estimate the effective population size of *P. vivax* parasites in PNG as 30,353 (13,043 – 69,142). It must be noted that the effective population size of parasite genetics is not the same thing as the effective population size for transmission in humans. Nevertheless, they are of the same order of magnitude.

Reviewer #2

This is a clear, well-written paper, and the conclusions are supported by the modeling results. The paper has practical deliverables that will be of use to the National Malaria Control Programme of PNG. This work should be published but it belongs more in Nature Scientific Reports than Nature Communications. Using mathematical models, the paper confirms well-known and expected results in malaria epidemiology. The quantitative guidance is of use to the PNG NMCP, but the paper does not present a major advance in malaria epidemiology or mathematical modeling.

This paper represents a significant advance that merits publications in Nature Communications for several reasons:

- (i) Malaria epidemiology is often conflated with *P. falciparum* epidemiology – the results presented here provide an important advance into the field of *P. vivax* epidemiology. While a great deal of *P. vivax* epidemiology is similar to *P. falciparum* epidemiology, there are many key differences, most notably related to relapses and the challenges associated with providing testing and treatment with primaquine or tafenoquine.
- (ii) This study highlights the important, previously unpublished but increasingly recognised result that combinations of existing malaria control interventions (including tafenoquine) will not be sufficient to eliminate *P. vivax* transmission in some regions.
- (iii) This manuscript describes the first individual-based model of *P. vivax* transmission capable of assessing the potential impact of malaria control interventions – a valuable tool for epidemiologists designing new intervention strategies.

1. Abstract - increased bed net coverage is predicted to reduce transmission but rebounds will occur if interventions are removed. This is an expected and well-understood result. The last sentence of the abstract aims to say something about future options for vivax control, but it is very non-specific.

We have now added to the end of the abstract to indicate how this model will be used by our group and others to simulate the design of future intervention trials.

2. The authors should make clear what parts of the individual-based model (IBM) were developed for this paper, and what parts were from the original work (ref 24). Also, it does seem appear that simulations for some of the figures may have been run with a more straightforward ODE model (maybe equations S1) as there is very little stochasticity in Figures 4 and 6. I can't be sure, because the figure captions are missing for the figures in the main text.

The individual-based model has been newly developed and this paper is the first place that it has been presented. However this model does build upon more theoretical insights from previous publications (White eLife 2014; White Proc Roy Soc B 2016). Additional text has been added to the manuscript to clarify this point.

The simulations in Figures 4 and 6 are run with the IBM. In the new simulations (see response to next point) the stochasticity is captured by presenting the results of multiple simulations. The

presentation of the stochastic variation is provided in Supplementary Figures S11-S19.

3. A major limitation is that the IBM runs are done with 20,000 individuals and replicated 3 times. For a Nature Communications publication, you should have the full PNG population size, some appropriate spatial resolution (province-level is probably fine here), and you should be doing 50 or 100 runs per scenario so that you can look at variation in hard-to-pin-down processes like rebounds. The rebound time will not always be the same in individual-based model runs.

This is a good suggestion that we have implemented. For each of the PNG provinces the population was estimated using data from the 2011 PNG census (see Supplementary Table S10). We increased the simulated population size from 20,000 to 100,000, with the exception of Manus which has a population of 60,485. The number of repeat simulations was increased from 3 to 100. The results of the IBM are now presented as the median with 95% range of the simulated output (see Supplementary Figures S11-S11).

4. It is good that an adherence parameter is included in the model, but a sensitivity analysis should be run here.

Our initial parameter estimate for LLIN adherence corresponded to a half-life for LLIN usage of 3 years, based on recommended values by the Roll Back Malaria Harmonization Working Group. In response to the reviewer's suggestion to undertake a sensitivity analysis, and another comment from reviewer 1, we investigated this parameter in more detail. In particular we were able to obtain data on the age distribution of Papua New Guinean nets (Katusele *et al*), from which we were able to estimate that the half-life of nets in PNG was 19.5 months, shorter than that of 23 (20-28) months estimated by Bhatt *et al* based on African data.

All simulations were therefore repeated assuming an LLIN half-life of 19.5 months. As sensitivity analyses, the results of simulations with an LLIN half-life of 23 months and 36 months (3 years) are also presented in Supplementary Figures S9 and S10.

RBM-HWG. 2014. Malaria implementation guidance in support of the preparation of concept notes for the global fund.

Katusele M, Gideon G, Thomsen EK, Siba PM, Hetzel MW, Reimer LJ. Long-lasting insecticidal nets remain efficacious after five years of use in Papua New Guinea. *PNG Med J* 2014; 57(1-4): 86-93

5. Model validation paragraph (second paragraph of results): this is excellent. Very important to acknowledge that with many spatial parameters overfitting will occur. Great work introducing this extra out-of-sample validation. This is rarely done with IBMs.

Thanks for this encouraging comment. It is our aim to demonstrate that the model captures some of the underlying realities of the *P. vivax* epidemiology across the globe, and is not just over-fitted to our epidemiological training data sets.

6. Since you have prevalence data for both 2010 and 2014 in PNG, you can also do an extra validation to see if the you get close to the 2014 final condition when you start with 2010 as an initial condition. Do you have enough information on interventions, drug coverage, bed net coverage, etc. from 2010 to 2014?

This is actually what we did for the simulations presented in Figure 4. We have adjusted the text to clarify this point.

7. I would remove the Thailand and Brazil simulations in Figure 7. This has the potential to be misused if picked up by the wrong person, and you haven't done all the background legwork, data assembly, descriptive epidemiology, spatial analysis that you have done for PNG. Figure 7 just shows general country level prevalence trajectories, under five scenarios, with no uncertainty shown.

We agree with the reviewer (and reviewer 3) on this point and the Thai and Brazilian simulations have been removed.

8. No movement between provinces in PNG. Again, for a Nat Commun paper, you would need to include this, even if it's a sensitivity analysis on a poorly known migration/movement rate.

We have made substantial efforts to estimate movement patterns between provinces in PNG. This is described in detail in the new Supplementary Section 4. In brief, we downloaded the flight schedules for the two airlines responsible for all, non-private, internal flights in PNG: Air Niugini and PNG Air. We used these data to estimate average daily air travel between provinces. We utilised a map of PNG roads and local knowledge of boat connections between island provinces to estimate inter-provincial travel by land and sea. A map of the estimated inter-provincial travel patterns is provided in Supplementary Figure S7.

The individual-based model was extended to simultaneously simulate populations in multiple provinces, allowing for daily travel between provinces. The code for this has been included on GitHub.

9. You mention a handful of times that the cohort of children born post-2009 may have low immunity after an intervention, and that they will make up the major part of the rebound if interventions are relaxed. Why not design an intervention specifically for this cohort? This would be a novel and as yet unstudied intervention type.

This is a sensible and pragmatic suggestion, and certainly reflects our line of thinking. Fully addressing this point would go beyond the scope of what we are aiming to achieve with this manuscript, but it is an area that our team are actively pursuing with a range of epidemiological studies in Papua New Guinea. The obvious LLIN-based strategies that can be modelled for these target groups are distribution of nets through ante-natal clinics and distribution to school age children.

Reviewer #3

1. The results are somewhat dispiriting and similar to prior results. The finding that MDA and vector control is not enough to eliminate malaria is not particularly different from past analyses, perhaps it would be useful/instructive to find potential solutions within the model (i.e., under what scenarios can elimination be achieved) and then see if those are possible to implement in real life.

We agree that the results are somewhat dispiriting, but they do reflect the difficult reality of attempting to control and eliminate malaria in *P. vivax* endemic countries with existing intervention strategies. While some of the results are similar to previous findings from the epidemiology of *P. falciparum*, it is important to stress that our analysis is focussed on the epidemiology of *P. vivax* which differs in several key ways, most notably the presence of relapses and the challenges associated with treatment.

In this analysis we have explored the potential impact of a wide range of combinations of interventions, but we did not choose to simulate the impact of interventions strategies that are not yet available (with the exception of tafenoquine which has recently completed phase 3 trials). The point about seeing what solutions are possible to implement in real life is a good one, and we see this model as a key component of this aim. In particular, we are using this model to aid the design of intervention strategies that our team are implementing in real life. For example, in Institut Pasteur we are coordinating an International Center of Excellence for Malaria Research (ICEMR) program where we are conducting epidemiological studies in Papua New Guinea and Cambodia to investigate the impact of new spatially targeted treatment and surveillance strategies. This model is allowing us to simulate a wide range of study designs, thus optimising our chances of success. Therefore, I think we're on the same page as the reviewer, but hope they will agree with us that designing new solutions is better done as part of a wider programmatic effort than in a single paper. We have taken the spirit of this suggestion on board and adjusted the abstract and discussion accordingly.

2. This paper I feel is a good demonstration of the problems in publishing today. The paper itself is light on details, while the supplementary materials reads like a thesis. I mean the supplementary material is 40 pages long and around 15,000 words. It would be much better to reduce extraneous components of the paper and somehow reduce the supplementary material. Perhaps by publishing the model as a separate paper in a different journal.

We have strived to write an intuitive and data-driven manuscript that is accessible to a wide range of individuals in the field, and not just experts in mathematical modelling. Coupled to this we have provided detailed Supplementary Information that facilitates the level of rigour and transparency, as well as providing the multiple additional sensitivity analyses and supplementary results that are increasingly demanded. On this occasion, we believe that the combined manuscript and Supplementary Information is the optimal way to present the work.

3. Despite the lengthy supplementary text, a major component of the model development is

completely glossed over: the model results described in the paper are based on a stochastic individual-based formulation of the model, however the development of this individual-based model is not described anywhere. This should be fully detailed as to the choices made on timing, how the code was translated, how the numbers of individuals were chosen (e.g., why such a small number), and how individuals were allocated between provinces. In addition, a full explanation should be made regarding why the authors chose to use a stochastic model implementation of a deterministic model. What was the benefit? Why go through this trouble? It is not clear to me why this would be the case. Lastly, for such a complex model, the code for producing the model should be made available as part of the project. Preferably on a code-hosting site such as bitbucket/github. In this day in age, it is not really appropriate to create complex mathematical models without also posting the code.

The reviewer makes very good suggestions here about (i) providing a description of the individual-based model; and (ii) publishing the code on GitHub. In brief, we have addressed both of these points.

The stochastic individual-based model has now been described in detail in a new Section 1.10 of the Supplementary Information. This describes in detail issues such as time steps and scheduling of events. The major advantage of using an individual-based model over a deterministic model is the ability to account for additional complexity and combinations of interventions. In the deterministic, compartmental framework the number of compartments grows extraordinarily large as the level of complexity increases. This point has now also been noted in the main text.

The code (including detailed comments) has now been uploaded to GitHub and is available for download from @MWhite-InstitutPasteur.

4. Despite the attention to detail in the model, the major assumptions greatly reduce its utility. The lack of mixing between provinces and the free mixing within provinces seem to be major issues that really hinder this model. Heterogenous biting and geographic distances from water sources have long been shown to be important in *P. falciparum* and there is no expectation that this would differ for *P. vivax*. It was actually quite disappointing to read how seemingly far behind this model was in terms of those aspects. Realistically, a simple model would have come to nearly the exact same conclusions, so again I am just totally mystified by the decision to make this an individual-based model. I could understand it if that resulted in some novel interpretation or provided additional value regarding how to implement interventions or included multiple things like heterogeneous biting or geographic variation in biting, but really the result seems that it would be the same if a much simpler model was used.

We have made substantial efforts to account for mixing between provinces by estimating inter-provincial movement patterns by air (downloads of Air Niugini and PNG Air flight schedules), road (obtaining maps of all roads in PNG), and sea (utilising local knowledge on boat routes). The individual-based model was extended to simultaneously model transmission in all provinces with inter-provincial travel. The results of these simulations are presented in Supplementary Section 4.

Heterogeneity in exposure to mosquito bites is explicitly accounted for in the model: it is assumed that the number of mosquito bites per person follows a log-Normal distribution (see Supplementary Information Section 1.3). The reviewer notes that distance from water sources has long been shown to be associated with malaria, and this is indeed correct although this relationship varies from place to place and between mosquito species. In the Papua New Guinean context, this is probably only important for *An. farauti* which favours large brackish water breeding sites (e.g. blocked river mouths). *An. punctulatus* favours small breeding sites that are common throughout all PNG villages

(e.g. car wheel ruts, pig wallows). Implementing this in the model would require a great deal of assumptions to be made in areas where we have little to no data.

Several of the conclusions of our analysis could indeed be replicated using a simpler model (e.g. using compartmental differential equation models). However, the key processes that are very challenging to implement in simpler compartmental models are LLIN distribution and loss of adherence, tracking combinations of interventions, and levels of population-level immunity. We have added to the methods section to highlight the advantage of using individual-based models, as well citing the recent review by The malERA Refresh Consultative Panel on Combination Interventions and Modelling where these issues are discussed in much more detail.

5. “In particular, we do not account for spatial variation in transmission intensity within each province which is known to vary substantially. Ideally we would be able to model transmission at the lower levels of Districts or Local Level Government (LLG) area. However, there is not sufficient data to meaningfully model malaria transmission at any spatial scale lower than the province level. “ spatial data is readily available these days, so breeding sites and thus transmission could be estimated. Even if not exact, how do these factors affect the conclusions. One would expect that heterogeneity could either improve or hurt elimination efforts and that it would be good to use mathematical models to estimate similar levels of incidence for different heterogeneities in transmission and see the differential effects of interventions.

There is much spatial data available these days, but in many developing countries there is little ‘actionable’ malaria data that can inform control and elimination efforts in real time. Fortunately, this situation is changing (slowly) in Papua New Guinea (see Rosewell *et al*), and our model is poised to take advantage of these developments when they eventually arise. We have added to the discussion to note this important point.

Rosewell *et al*. Health information system strengthening and malaria elimination in Papua New Guinea.

6. The Figure Legends are numbered incorrectly

We apologise for this oversight. This has now been addressed.

7. One major assumption is that implementation of MDA could even proceed without G6PD testing. This is mentioned in passing, but seems that should be a larger focus of the sensitivity analysis.

We have assumed throughout that MDA with primaquine or tafenoquine is implemented with screening for G6PD deficiency. We have highlighted this assumption in both the text and Supplementary Information.

8. The paper is based around PG, cut all the extra stuff about Thailand and Brazil from the paper. Though Figure 3 does suggest that the model gets the general gist of the data, it clearly is not capturing the wide variation across the world, and does not provide a ton of confidence that this can explain global variations. It also just makes a sprawling unwieldy paper less focused. I think this paper should just clearly focus on the issues of PG, and then perhaps in the discussion mention note that it could perhaps be applied in other settings if the data were collected to provide that avenue – that is all that is really necessary.

We have acted on the reviewer's concern and removed the section about Thailand and Brazil, and noted in the discussion that this model can be applied in other settings.

There is indeed wide variation across the world, as evidenced by the data used for model validation in Figure 3. While the model captures the general pattern of the data (even the noisy clinical incidence data in Figure 3a), we cannot expect to obtain a more detailed fit without additional data on age ranges and relapse phenotype for each of the study sites represented by the data points.

Figure 2: What is the deal with East Sepik, why are there no dots in the older ages?

The study in East Sepik focussed only on children aged 0.8 to 3.2 years of age. The continuation of the lines above these age groups in Figure 2 represents the model prediction in older ages.

Figure S3: To my eye the line does not fit the data very well. Why not use something like STL to fit the data? Or perhaps something that is less perfectly regular? Since from the data it appears that seasonality is moving around a bit, how would that affect the outcomes, would it have any affect? Also it is noted that equation S17 has seasonal equation, but it is S18.

It is clear that the lines in Figure S3 do not capture all of the variation exhibited in the data. However, this is a consequence of bringing together a wide range of entomological data sets. There really is a lot of noise in mosquito catch numbers, and although there are clear seasonal patterns, they are very hard to accurately predict from season to season. We have now calculated and plotted the confidence intervals for the fits portrayed in Figure S3. We have also explored the suggestion of using seasonal trend decomposition using loess (STL), with the resulting seasonal pattern shown via the red lines in the Additional Figure 2 below. Reassuringly, in all three cases the red lines fall almost entirely within the 95% confidence intervals estimated using our smooth functional form.

We have also further tested how variation in seasonal mosquito patterns will lead to variation in simulate *P. vivax* prevalence. The results of this are shown in Additional Figure 1 in the response to Reviewer 1.

Additional Figure 2: Data on seasonal abundance of three species of adult female Anophelines from villages in Papua New Guinea and the Solomon Islands. The black lines show the fitted functional form and the shaded grey region shows the 95% confidence interval. The red line shows the seasonal form predicted by seasonal trend decomposition using loess (STL).

Reviewers' comments:

Reviewer #2 (Remarks to the Author):

I realize the authors have put a lot of work into the revision. I still think the paper tackles a very straightforward question: will more frequent LLIN distribution or higher LLIN coverage reduce malaria prevalence? The answer is yes. And I still have some questions about the construction of their individual-based model. I do not think this is an appropriate manuscript for Nature Communications.

My general comments are:

1. The authors claim that this is the very first individual-based model of vivax. It is true that most of the major modeling groups have not spent the majority of their time on building agent-based models of Plasmodium vivax.

- The Swiss TPH group appears to have a P. vivax model that is out-of-date and being updated.
- An Ecuadorian group has published two papers using their P. vivax agent-based model

<http://journals.plos.org/plosone/article?id=10.1371/journal.pone.0193493>
<https://malariajournal.biomedcentral.com/articles/10.1186/s12936-015-1030-7>

- The IDM group do not appear to have a P. vivax module in EMod
- The Oxford groups do not appear to have P. vivax transmission in any of their published models

2. The authors' individual-based model does not seem to be very stochastic. The figures showing prevalence trajectories are just single lines, unless the 95% ranges are so narrow that they look like single lines. In which case, the model shows almost no variation among the different simulation runs. Something is wrong here. Especially, when the simulations reach 1% prevalence and 0.1% prevalence. There should be substantial variation in this range as different simulations will go extinct at different times.

3. I do not understand the comment about rebounds being shown in figure S12. Are there really 100 simulations in each panel here with the rebound occurring at the same time for each trajectory? Again, this does not seem right for an individual-based model. The rebound time should be more variable than this. In a standard waiting process, the variance is the square of the mean.

4. Fig 4 and Fig 6 seem very smooth, especially at the beginning of the trajectory before any interventions are introduced. (maybe the authors put the times series through a smoothing function before plotting it?). Are these really simulations from a stochastic individual-based model? How long were they run before the year 2000 so they could reach

this stationary equilibrium behavior?

Reviewer #3 (Remarks to the Author):

I appreciate the efforts of the authors to respond to all the comments and for posting the code online. I have no additional comments and look forward to the publication of this important paper.

Reviewer #2

I realize the authors have put a lot of work into the revision. I still think the paper tackles a very straightforward question: will more frequent LLIN distribution or higher LLIN coverage reduce malaria prevalence? The answer is yes. And I still have some questions about the construction of their individual-based model. I do not think this is an appropriate manuscript for Nature Communications.

My general comments are:

1. The authors claim that this is the very first individual-based model of vivax. It is true that most of the major modeling groups have not spent the majority of their time on building agent-based models of Plasmodium vivax.

- The Swiss TPH group appears to have a P. vivax model that is out-of-date and being updated.
- An Ecuadorian group has published two papers using their P. vivax agent-based model

<http://journals.plos.org/plosone/article?id=10.1371/journal.pone.0193493>

<https://malariajournal.biomedcentral.com/articles/10.1186/s12936-015-1030-7>

- The IDM group do not appear to have a P. vivax module in EMod
- The Oxford groups do not appear to have P. vivax transmission in any of their published models

We thank the reviewer for bringing the work of the Ecuadorian group to our attention. These papers have now been cited in the introduction.

To our knowledge, the major malaria modelling groups have not published anything on P. vivax. While we have heard of ongoing work on P. vivax modelling at Swiss TPH, we have not seen any work published or presented at international conferences.

2. The authors' individual-based model does not seem to be very stochastic. The figures showing prevalence trajectories are just single lines, unless the 95% ranges are so narrow that they look like single lines. In which case, the model shows almost no variation among the different simulation runs. Something is wrong here. Especially, when the simulations reach 1% prevalence and 0.1% prevalence. There should be substantial variation in this range as different simulations will go extinct at different times.

The individual-based model is indeed stochastic, but the large population size means that all the stochastic simulations very closely track the equivalent results of a deterministic model. This point has been noted in the methods sections. At this stage, it is worth noting that the simulated population size was increased from 20,000 to 100,000 in response to a request from Reviewer #2's previous review.

The result that stochastic models closely approximate deterministic models for large population sizes or large values of R_0 is, in general, a well understood textbook result. See for example:

Keeling MJ, Rohani P. Modelling infectious diseases in humans and animals (Chapter 6). *Princeton University Press*

Grenfell BT, Bjornstad ON, Finkenstadt BF. Dynamics of measles epidemics: scaling noise, determinism, and predictability with the TSIR model. *Ecological Monographs* 2002; 72(2): 185-202

3. I do not understand the comment about rebounds being shown in figure S12. Are there really 100 simulations in each panel here with the rebound occurring at the same time for each trajectory? Again, this does not seem right for an individual-based model. The rebound time should be more variable than this. In a standard waiting process, the variance is the square of the mean.

There are 100 simulations in each panel. There is variation in the timing of the rebound. Consistent with the theoretical work cited in response to the previous point, this variation is most pronounced in the low transmission provinces (Western, Gulf & Central). To further illustrate this point we have generated new simulations with a substantially smaller population size of 1,000 where we see much greater variation (Supplementary Figure S13).

4. Fig 4 and Fig 6 seem very smooth, especially at the beginning of the trajectory before any interventions are introduced. (maybe the authors put the times series through a smoothing function before plotting it?). Are these really simulations from a stochastic individual-based model? How long were they run before the year 2000 so they could reach this stationary equilibrium behavior?

The curves in Figure 4 and Figure 6 are smooth because of the effect of large population sizes substantially reducing stochastic deviations from the mean as discussed above. There was no smoothing of time series. The individual-based model was initialised using the equilibrium solution of the non-seasonal deterministic model in 1990. This point has now been added to the methods.

Reviewer #3

I appreciate the efforts of the authors to respond to all the comments and for posting the code online. I have no additional comments and look forward to the publication of this important paper.

We wish to thank the reviewer for their supportive comments, and we're glad they appreciate our steps at publishing code online.